# Stimulus domain transfer in recurrent models for large scale cortical population prediction on video

**Fabian H. Sinz,[1-2,5,7,*] Alexander S. Ecker,[2,4-6] Paul G. Fahey,[1-2] Edgar Y. Walker,[1-2]**
**Erick Cobos, [1-2] Emmanouil Froudarakis,[1-2] Dimitri Yatsenko,[1-2]**
**Xaq Pitkow,[1-3] Jacob Reimer,[1-2] Andreas S. Tolias[1-3,5]**

[1] Department of Neuroscience, Baylor College of Medicine, Houston, TX, USA
[2] Center for Neuroscience and Artificial Intelligence, Baylor College of Medicine, Houston, TX, USA
[3] Department of Electrical and Computer Engineering, Rice University, Houston, TX, USA
[4] Centre for Integrative Neuroscience, University of Tübingen, Germany
[5] Bernstein Center for Computational Neuroscience, University of Tübingen, Germany
[6] Institute for Theoretical Physics, University of Tübingen, Germany
[7] Institute for Computer Science, University of Tübingen, Germany

[*]`sinz@bcm.edu`

## Abstract

To better understand the representations in visual cortex, we need to generate better predictions of neural activity in awake animals presented with their ecological input: natural video. Despite recent advances in models for static images, models for predicting responses to natural video are scarce and standard linear-nonlinear models perform poorly. We developed a new deep recurrent network architecture that predicts inferred spiking activity of thousands of mouse V1 neurons simultaneously recorded with two-photon microscopy, while accounting for confounding factors such as the animal's gaze position and brain state changes related to running state and pupil dilation. Powerful system identification models provide an opportunity to gain insight into cortical functions through *in silico* experiments that can subsequently be tested in the brain. However, in many cases this approach requires that the model is able to generalize to stimulus statistics that it was not trained on, such as band-limited noise and other parameterized stimuli. We investigated these domain transfer properties in our model and find that our model trained on natural images is able to correctly predict the orientation tuning of neurons in responses to artificial noise stimuli. Finally, we show that we can fully generalize from movies to noise and maintain high predictive performance on both stimulus domains by fine-tuning only the final layer's weights on a network otherwise trained on natural movies. The converse, however, is not true.

## 1 Introduction

The visual cortex represents natural stimuli in a complex and highly nonlinear way [1, 2]. In order to understand these representations, we need predictive models that can account for neural responses to natural movies. This task is particularly challenging because a substantial portion of the response variability in cortical neurons is not driven by the stimulus, but by other factors such as eye movements under free-viewing conditions and brain state changes [3–9]. While deep convolutional networks have recently been shown to improve prediction performance over linear-nonlinear type models [10–13] and are currently considered state-of-the-art, in V1 they have only been used to predict responses to

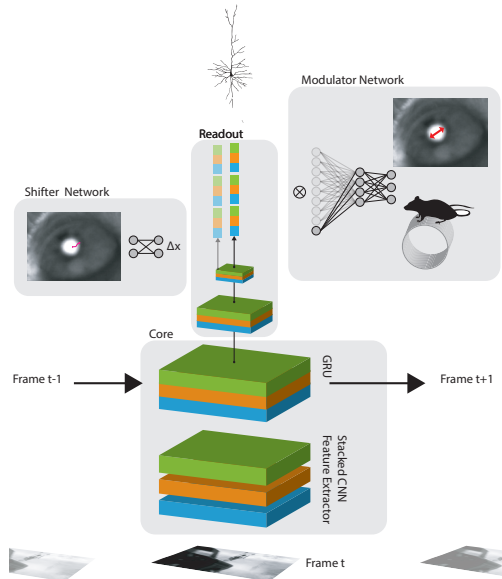

Figure 1: One timestep of the recurrent architecture: The network consists of a core, a readout, a shifter, and a modulator. The core uses a CNN to generate a non-linear feature representation of one frame of the movie, which is then fed into a GRU. The readout decomposes the hidden state of the GRU into different scales and reads from one spatial location per neuron across all features and scales using a spatial transformer grid point. The shifter predicts a shift for the entire neuron population based on the pupil position. The modulator predicts a gain term per time point and neuron based on the running state and pupil dilation of the animal.

static natural images. These models also make sub-optimal use of the data, because they currently do not account for stimulus-independent variability. Furthermore, system identification is only one step towards a better understanding of cortical representations. Successful models must be able to generalize beyond the stimulus statistics they are trained on to generate new insights that can be tested back in the brain. This domain transfer is a known hard problem in neural system identification and machine learning in general, since it requires generalization beyond the statistics of the training set.

We make three contributions towards this goal: (i) we propose a novel recurrent neural network architecture that can simultaneously predict the responses of thousands of cortical neurons while accounting for neural variability caused by eye movements or brain state changes related to measurable factors such as running state or pupil dilations; (ii) we demonstrate that training our model on natural movies allows some extent of domain transfer and recovers neurons' tuning properties, such as orientation tuning, direction tuning, and receptive field structure, under artificial stimuli; and (iii) we analyze the limits of domain transfer to show that models trained directly on the target domain always outperform those trained on other domains in terms of predictive performance. However, we also demonstrate that the nonlinear feature representation learned on natural movies can transfer well to the noise domain by fine-tuning the last layer's weights, while the converse is not true (i. e. generalization from noise to movies). In fact, there exists a single set of feature weights that leads to optimal performance in both domains, even though we currently cannot identify this set of weights from natural movies alone.

## 2   Network Architecture

Our network consists of four components (Figure 1): a *core* providing nonlinear recurrent features from the video frames, a *readout* mapping the core features to each neuron's activity, a *shifter* predicting receptive field shifts from pupil position, and a *modulator* providing a gain factor for each neuron depending on the running state and the pupil dilation of the animal (Figure 1).

**Core**   The core consists of a three-layer, 2d convolutional neural network (CNN), applied separately to each frame, followed by a convolutional Gated Recurrent Unit (GRU) [14]. Each CNN layer is composed of a 2d convolution layer, a batch normalization layer [15], and an ELU nonlinearity [16]. The first layer has a skip connection into the third layer which is stacked onto the input. That is, if the first and second layer have $k_1$ and $k_2$ output channels, the third layer has $k_1 + k_2$ input channels. Finally, similar to DenseNets [17], the outputs of all layers are stacked and fed to a one-layer convolutional GRU that models the lag between input and neural response and recurrently computes nonlinear features. Previous work demonstrated that neural response properties like direction selectivity can be modeled by recurrent networks [18].

**Readout**    We model the neural response as an instantaneous affine function of the hidden state of the GRU at time $t$ followed by an ELU nonlinearity and an offset of 1 to make the response positive. At each point in time the hidden state of the GRU is a tensor $v \in \mathbb{R}^{w \times h \times c}$. The straightforward approach of using a fully connected layer $\sum_{ijk} w_{ijk} v_{ijk}$ [10] or a factorized layer $\sum_{ijk} w_{ij} w_k v_{ijk}$ over features and space [12] requires strong regularization to deal with the large number of parameters and can potentially lead to ghosting artifacts where the receptive field of a neuron shows up at several locations if the spatial components $w_{ij}$ are non-zero at more than one spatial location because of fitting noise or ambiguities in the data caused by gaze shifts. To circumvent these issues we explicitly incorporate the prior knowledge that a neuron only reads out from one particular location in space and model each neuron with a spatial transformer layer [19] with a single grid point reading from $v_{ijk}$. Thus, each neuron $i$ is parameterized by a relative spatial location $(x_i, y_i) \in [-1, 1]^2$ at which the spatial transformer layer extracts a local feature vector $v_{x_i y_i}$ by bilinearly interpolating the adjacent pixels; a linear-nonlinear function later combines these features into a single neural response $y_i = f(\mathbf{w}_i^\top \mathbf{v}_{x_i y_i:} + b_i)$. Since, a priori, we do not know the spatial location of the neuron, the $(x_i, y_i)$ become part of the network parameters and are learned via gradient descent. However, this introduces another problem: when the initial estimate of the location is far away from the neuron's actual location, there is little gradient information to nudge the grid point to the correct place. We therefore decompose $v_{ijk}$ into $\ell$ spatial scales through repeated application of a $p \times p$ average pooling layer with stride $p$ until the smaller spatial dimension is only one pixel in size: $\mathbf{v}^{(j)} = \text{pool}^{(j)}(\mathbf{v})$. The spatial transformer layer then extracts $\ell$ feature vectors from the same relative location $(x_i, y_i)$ at each scale and stacks them into a single feature vector of dimension $k \times \ell$ fed to the final affine function and nonlinearity $y_i = f(\sum_{j=1}^{\ell} \mathbf{w}_{ji}^\top \mathbf{v}_{x_i y_i:}^{(j)} + b_i)$ (Figure 1). Importantly, the relative spatial location is shared across scales.

**Shifter**    Unlike primates, mice are not trained to fixate their gaze in a single position, complicating eye tracking. To model the responses of thousands of neurons in a free viewing experiment, we take an alternative approach and directly estimate a receptive field shift for all neurons from the tracked pupil position solely based on optimizing the predictive performance of the network. Specifically, we feed the pupil location $\mathbf{p} \in \mathbb{R}^2$ into a network that predicts a shift $\Delta \mathbf{x} = (\Delta x, \Delta y) \in \mathbb{R}^2$ for each time point which is added onto all locations $\mathbf{x}_i = (x_i, y_i)$ of the spatial transformer readout. Note that the pupil location is measured in coordinates of the camera recording the eye, while the shift needs to be applied in monitor coordinates. This transformation can either be estimated by a calibration procedure [20–22], or learned from the data using regression on pairs of eye camera–monitor coordinates. Our approach differs from previous ones in that it estimates gaze shifts purely based on prediction performance. To a first approximation, the mapping from pupil coordinates to gaze shifts is affine. We therefore use a one layer perceptron (MLP) with a tanh nonlinearity for predicting $\Delta \mathbf{x}$ for all neurons. We empirically found that clipping $\mathbf{x}_i + \Delta \mathbf{x}$ back to $[-1, 1]^2$ improves the performance of the network.

**Modulator**    To account for fluctuations in neural responses unrelated to the visual stimulus, we use variables known to correlate with brain state—pupil dilations (and their derivative) and absolute running speed of the animal [4, 5, 23]—to predict, per timepoint, a neuron-specific multiplicative gain factor applied to the output of the readout layer. We use a GRU followed by a fully connected layer and an exponential nonlinearity offset by one to predict this factor and model the unknown delay between behavioral state and neural gain.

## 3  Related Work

There is a number of previous studies that predict neural responses to natural images or video, differing in the degree to which parts of the network are hand-crafted, the complexity of the network, the neural responses they are fitted to (electrophysiology vs. two-photon), the species the data was recorded in (mouse, cat, monkey), whether the animal was anaesthetized, and whether multiple neurons share parts of the network. None of the previous approaches predict a comparably large number of neurons, very few use video, and none simultaneously account for eye shifts and brain state modulations.

Gallant and colleagues were the first to fit models to movies of natural scenes predicting the responses in macaque area V1 and MT [1, 24–26]; their models are either spatio-temporal linear-nonlinear

models or use hand-crafted non-linear features such as power in the Fourier domain or divisive normalization. Since the monkey is fixating during their recordings, there is less need to consider eye movement and brain state. Lau et al. trained a multi-layer perceptron with inputs from different delayed time points to predict responses of V1 neurons to random bars [27]. Vintch et al. trained a two state linear-nonlinear model with a convolutional first layer to predict cell responses in monkey V1 [28]. Similar to our work, Batty et al. used a multi-layer recurrent network as a feature representation for a linear-nonlinear model to predict retinal ganglion cells [29]. Sussillo et al. used a variational autoencoder to infer latent low dimensional dynamics to explain neural responses [30]; their model was only tested on synthetically generated data. Other studies predict neural responses to static artificial or natural images [10–12, 26, 27, 31, 32]. Zipser and Andersen were one of the first to use neural networks to predict neural data; they used visual input and eye position to model the responses of neurons in area 7a whose neurons are involved in visuo-motor coordination [33].

## 4   Experiments

**Neural and Behavioral Data**   Our data consists of three sets of 1344-4692 simultaneously recorded deconvolved fluorescence traces [34] from two-photon scans in mouse visual cortex area V1 L2/3, collected from three animals using a large-field-of-view mesoscope [35]. Cells were selected based on spatial features of the segmented masks, but disregarding visual responsiveness. The acquisition frame rate was roughly 6Hz. Pupil position, dilation, and absolute running speed of the animal were monitored with an eye tracking camera and a styrofoam treadmill. The contour of the pupil was extracted semi-automatically for each frame. The center and the major radius of a fitted ellipse were used as the position and the dilation of the pupil. All behavioral traces were lowpass filtered to 2.5Hz using a hamming filter. To match the frame rate of the stimuli, all neural and behavioral traces were subsequently upsampled to 30Hz using linear interpolation. Data can be downloaded from `https://web.gin.g-node.org/cajal/Sinz2018_NIPS_data`.

**Stimuli**   The mice were presented with natural video (10s clips from both, Hollywood movies and rendered 3D scenes), and parametric noise clips on a standard LCD monitor. Noise movies consisted of ten minutes of bandpass filtered Gaussian noise with interleaved periods of drifting orientation bias and ten minutes of the cosine of a low spatial frequency Gaussian process. Real natural scenes included 42 min of 10s clips extracted from Hollywood action movies and the YouTube 1M dataset [36]. Rendered natural scenes consisted of 21 min of 10s clips produced using unreal engine with custom scenes and programmed camera flights. All movies were converted to grayscale and presented at 30Hz. Prior to feeding the data to the network, all frames were downsampled to $36 \times 64$px. Videos that did not match the 16:9 ratio were center cropped.

**Network Implementation**   All numerical experiments and analyses were performed using Data-Joint [37], Numpy/Scipy [38], Matplotlib [39], Seaborn [40], Jupyter [41], PyTorch [7], and Docker [42]. All models were trained on NVIDIA TitanX, 1080ti, or TitanV. Code is available from `https://github.com/sinzlab/Sinz2018_NIPS`. The core used $3 \times 3$ zero-padded convolutions (except for the first layer that used $7 \times 7$) with 12 features in each layer. First layer filters were regularized with an L2 norm on the Laplace filtered weights to encourage low frequency filters. The filters of the hidden layers were regularized using a group sparsity regularizer on all filters corresponding to one output channel. Batch normalization used a momentum term of $0.1$. Convolutional layers in the GRU used $3 \times 3$ zero-padded convolutions with 36 feature channels, and no regularization. The initial state of the GRU was learned as an additional parameter. Nonlinearities in the core were ELUs. The readout used five $4 \times 4$ average pooling steps with stride 4, and an ELU+1 nonlinearity to keep the neural responses positive. We also tried $4 \times 4$ with a stride of 2 but did not find a strong effect on the performance. The readout weight vectors were L1 regularized. The bias of the readout was initialized to match the mean response of the respective neuron. While developing the network we found that this speeds up optimization, but does not affect the final performance. The shifter used L2 regularization on the weight matrix. The GRU of the modulator used 50 hidden channels.

**Training Schedule and Hyper-Parameter Selection**   Due to the large number of hyper-parameters, the specific network and training settings were determined using a combination of grid search and manual exploration on a validation set. We selected for kernel size, channels, and regularization constants. We found that strong input filter regularization helps, and sparse regularization in the

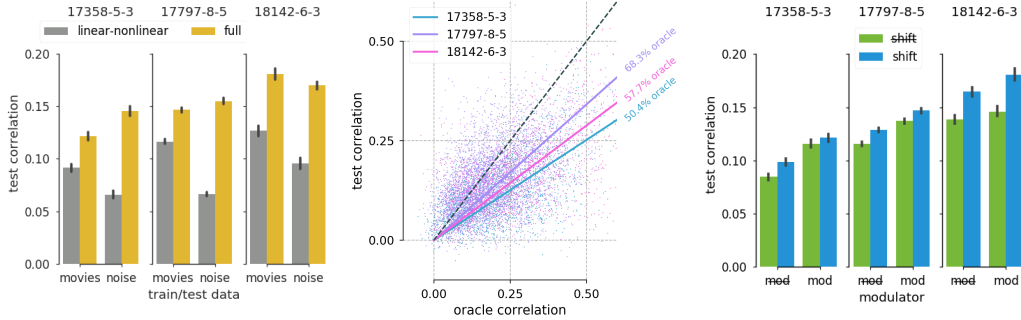

(a) full model vs. linear-nonlinear  (b) oracle vs. model performance  (c) influence of shifter and modulator

Figure 2: Performance of the model. (a) Single trial correlation of the model prediction with the neuronal responses across neurons for the full recurrent model and a spatio-temporal linear-nonlinear model equipped with a shifter and modulator. Error bars mark 95% confidence interval. (b) Model test correlation against oracle correlation, which represents an upper bound of the achievable performance bases solely on the stimulus. The percentage of oracle was computed from the slope of a linear regression without offset. (c) Influence of the single network components on the prediction performance.

readout is important. When the number of channels is too large, we run into overfitting problems. Within a reasonable range of kernel sizes the network is quite insensitive to the kernel size. Afterwards, we used the same settings for all recordings, unless we explicitly explored the effect of different network architectures on performance. Networks were either trained on noise movies or natural movies. Each training batch contained 8 clips of 5s each, randomly selected from longer clips in the training set. Validation and test scores were computed on the full length clips. The pupil position was standardized to mean zero and standard deviation one across the entire training set. The behavioral traces were divided by their standard deviation. We used time-averaged Poisson loss $\langle \hat{y} - y \log(\hat{y} + \varepsilon) \rangle_{t>t_0}$ to train the models, with $\varepsilon = 10^{-16}$ for numerical stability and an initial burn-in period of $t_0 = 15$ frames to allow the recurrent networks to settle in from the initial state. We optimized the objective using ADAM [43] and a two stage training schedule with step size $0.005$ and $0.001$, respectively. Each training stage finished if either the correlation between single trials of the validation set and the model responses (using the same burn in period of 15 frames) did not improve over the current best result for 5 occasions checked every 4 sweeps through the training set or if the number of sweeps through the dataset exceeded 500. At the end of each training stage, the model was reset to the best performing model within that stage.

## 5 Results

### 5.1 Performance

We trained networks on noise or natural movies from scans on three different animals (17358-5-3, 17797-8-5, 18142-6-3). We measured the performance by the correlation between the model prediction and the neural responses across six 10s test clips each repeated 10 times. The networks reach an average correlation across neurons of $0.145$ to $0.18$, depending on the particular scan and movie type. Note that these are single-trial predictions at 30 Hz, thus the relatively low prediction accuracy; we will discuss this below. We compared our recurrent network to a linear-nonlinear model consisting of a 3D convolutional layer with filter size 13 (in space and time) and 36 channels, a batch norm layer, and the same readout, shifter, and modulator architecture as the recurrent model to allow it to account for variability unrelated to visual stimulus. This model had a comparable number of parameters as our network. Ignoring components common to both and biases the number of parameters were: linear-nonlinear model $13^3 \cdot 36 = 79092$ (3D-conv); ours 91740 total parameters ($7^2 \cdot 12 + (12^2 + 24 \cdot 12) \cdot 3^2$ in CNN, $6 \cdot 3^3 \cdot 36^2$ in GRU)). However, even though the linear-nonlinear network had on the same order of parameter and used the same shifter or modulator components, our recurrent network consistently performed better (Figure 2a).

Cortical neurons naturally exhibit a substantial degree of variability in their responses which affects their predictability. To get an idea of the model performance relative to the best achievable performance, we correlated each trial in the test set with the "oracle" estimator, computed by correlating the mean over the $n - 1$ other repeats with the remaining trial, and averaging that over all splits and repeated images. We estimated the percentage-of-oracle as $100\times$ the slope of a linear regression without offset fitted to the oracle and model test correlations. All networks achieve 50% to 70% of the achievable oracle score (Figure 2b). These scores are for natural movies only, since noise movies were not repeated. Note that the network performance could in principle be better than the oracle performance, since the oracle is only computed on repeats of the stimulus and not on trial specific-behavioral variables to which the network has access. In order to measure the contribution of the shifter and modulator components on the prediction performance, we trained networks with those components turned off. Without shifter or modulator, the percent oracle scores were 35.7% (17358-5-3, 50.4% with), 56.3% (17797-8-5, 68.3% with), 44.5% (18142-6-3, 57.7% with). This showed that both components improve the network performance (Figure 2c). The relative contribution of each component depends on the particular dataset.

# 6 Domain transfer

In the following two sections, we explore to what extent a model trained on natural videos can predict neural responses and tuning properties determined by noise stimuli.

## 6.1 Tuning

We first mapped receptive fields of the networks trained on either natural movies or noise with a newly generated set of colored noise and compared them to receptive fields of neurons mapped with reverse correlation. Figure 3a shows a selection of receptive fields for neurons with the best prediction scores on natural movies, along with the receptive field of the real neuron and the model trained on noise movies. Qualitatively, the orientation, location, and general sub-field structure matches between the networks and neurons.

Next, we computed direction tuning curves for the real neurons and their respective model neurons in models trained on noise and natural movies (Figure 3b). On average, both models correctly infer the preferred orientation, but sometimes exhibit a sign flip in the direction. The model trained on noise typically exhibits a closer match with the tuning of the real neuron. We quantified this by computing the distribution of the difference $\Delta\phi$ in preferred orientation between model neurons and their real counterparts. We considered all neurons whose direction tuning functions had an $R^2 > 0.005$ and an orientation selectivity index OSI$> 0.2$ ($R^2 > 0.002$ and DSI$> 0.1$ for direction selectivity; {D,O}SI$= (r_p - r_a)/(r_p + r_a)$ for $r_p, r_a$ are the mean responses in the preferred and anti-preferred orientation/direction), and models trained on noise and natural movies, as well as with and without shifter and modulator networks (Figure 3c). In all instances, the distributions are centered around zero which means that all the models predict the correct orientation on average. However, for orientation selectivity models on natural movies without shifter and modulator components exhibit a larger variance ($p < 0.03$, $p < 0.0015$, $p < 10^{-11}$ for the three scans using Levene' test) and slight biases in the median of the distribution. This indicates that accounting for confounding variables can be relevant in domain transfer for neural prediction. The models trained on noise exhibit a substantially lower variance in $\Delta\phi$. A similar pattern is seen when quantifying the difference in preferred direction (Figure 3d).

## 6.2 Limits of domain transfer

While the network trained on natural images generally predicts the correct tuning properties of the neurons, there is a clear drop in the quality of tuning property prediction across stimulus domains.

One possible reason for this could be that a network core trained on natural movies does not provide the right features to predict responses to noise. To test this hypothesis, we trained a network with three readouts all referring to the same neurons and a core exclusively trained on one stimulus domain (Figure 4a). To ensure that the core was only trained on one stimulus domain, we stopped the gradient of both other readouts before the core. One readout was trained with natural movies only, one with

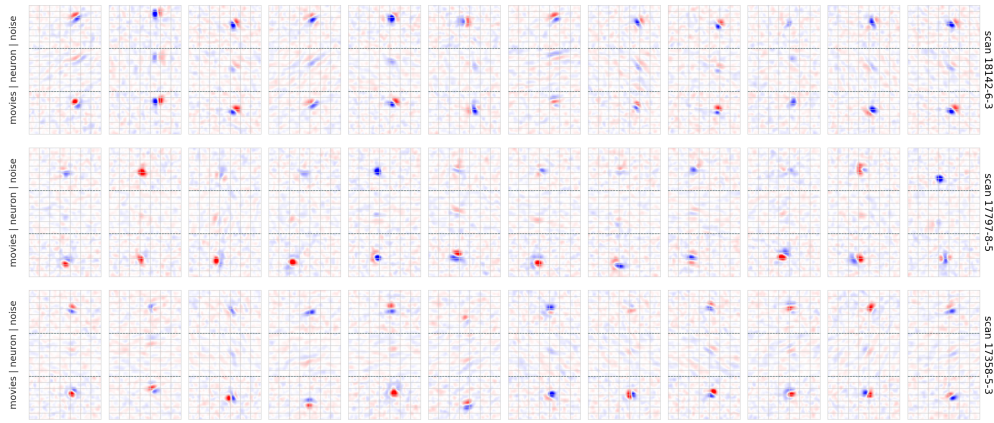

(a) Receptive fields

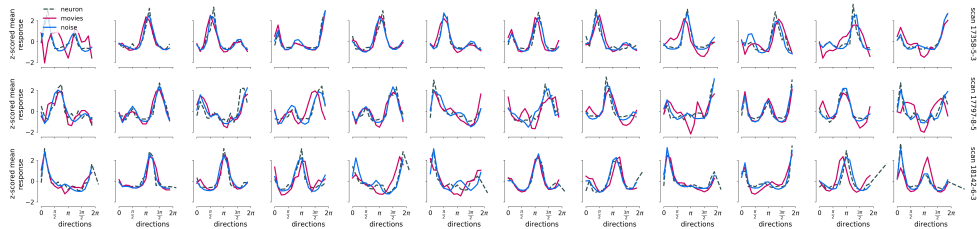

(b) Direction tuning curves

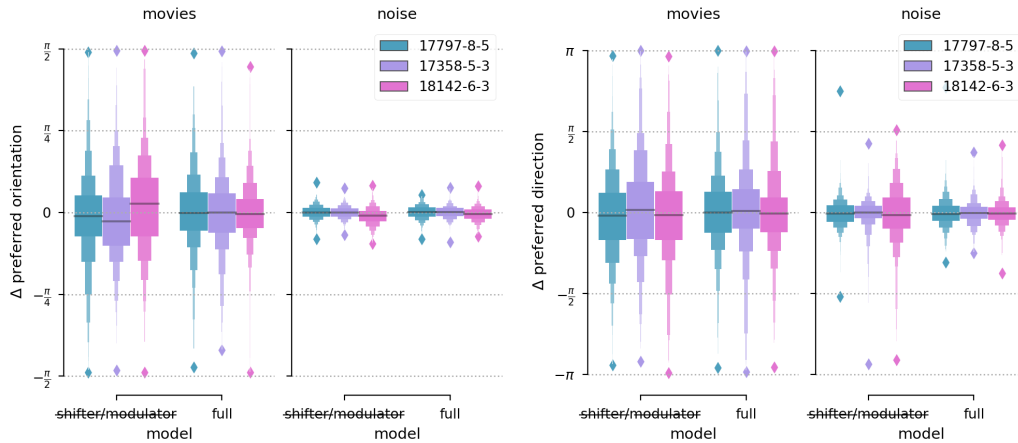

(c) Difference in preferred orientation.

(d) Difference in preferred direction.

Figure 3: (a) Receptive fields computed by reverse correlation of colored Gaussian noise with responses of a network trained on neural responses to noise (top row in each group), the actual neural responses (middle row), and a network trained on responses to natural video. For presentation, we averaged the spatio-temporal receptive field over the first 300ms. (b) Direction tuning curves of real neurons (dashed) and their model counterparts trained on natural movies (pink) and (noise) blue. We show curves for neurons that were best predicted on natural images, among all neurons that exhibited direction tuning with $R^2 > 0.005$ and an orientation selectivity index (OSI) $> 0.2$. Curves are z-scored to match the scale. (c-d) Difference in preferred orientation (left) and direction (right) between neurons and model neurons trained on either natural or noise movies for model with shifter and modulators, and without. We only considered neurons with $R^2 > 0.005$ and OSI$> 0.2$ ($R^2 > 0.002$ and DSI$> 0.1$ for direction selectivity). Lines denote the median.

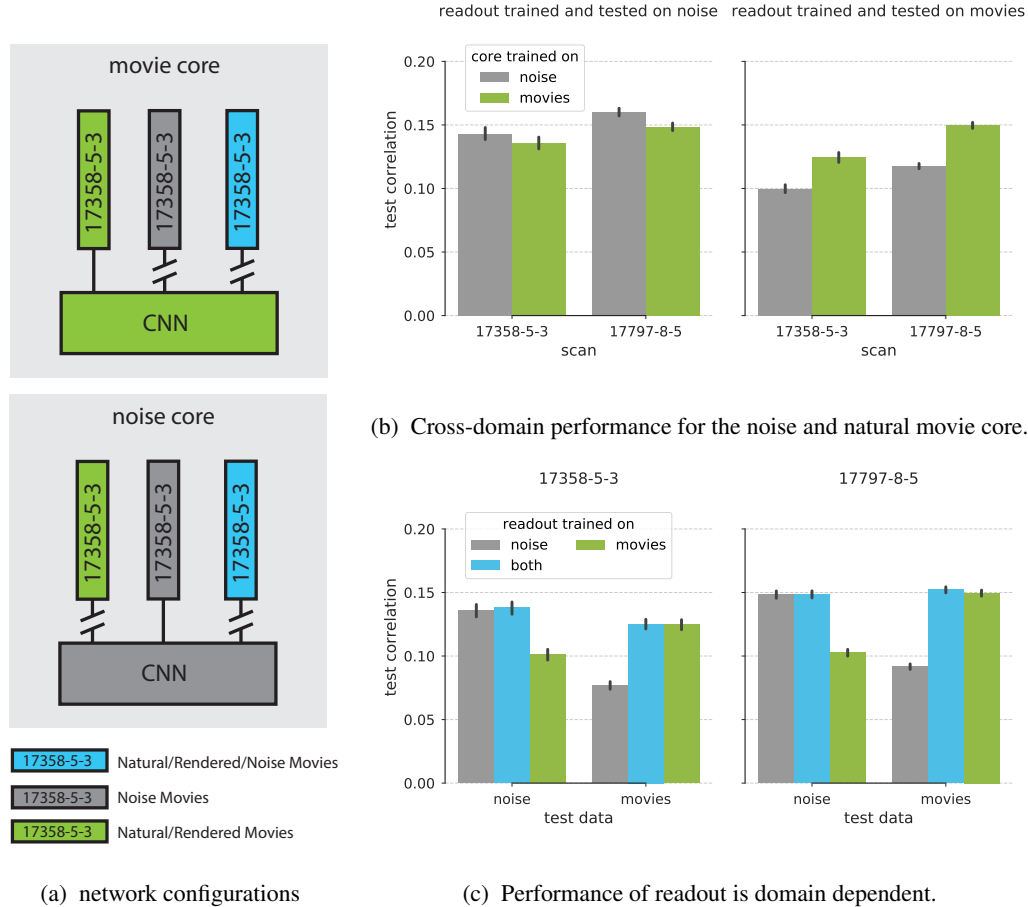

(a) network configurations

(b) Cross-domain performance for the noise and natural movie core.

(c) Performance of readout is domain dependent.

Figure 4: (a) Network configuration for domain transfer training. Vertical squares represent readouts. Interrupted lines denotes gradient stopping during training. All readouts are trained on the same set of neurons but on different stimulus domains. (b) The core trained on natural movies performs well on noise if the readout is trained on noise (left panel), but not vice versa (right panel). (c) Using the natural movie core, performance drops when the readout is not trained on the respective domain. However, one readout trained on both domains performs as well as each readout trained dedicated on the test domain. Note that the left and right bar groups in each panel refer to the same network.

noise movies only, and one with both. We also shared the grid locations of the neurons between the readouts, and balanced the stimulus domains in the batches for the shared readout.

We first compared the performance of each core (trained on natural movies vs. noise) always using the readout trained on the target domain (i. e. readout trained on the same stimulus domain as the test set). We found that both the natural movie core and the noise core perform well when testing on noise. In contrast, we observe a substantial performance drop for the noise core when testing on natural movies (Figure 4b). This result shows that the feature representations of the natural movie core are rich enough to transfer to noise stimuli, while the converse is not true. One possible confounding factor for this finding is the different stimulus presentation times between noise (20min) and natural movies (∼1h). We ran a control experiment to show that this is not the case (see supplementary material). We also ran a control experiment to show that there is no drop in performance between networks trained on rendered movies vs. natural movies only (see supplementary material).

We next turn to the setting where the readout is not trained without using stimuli from the target domain. For this experiment we always use the core trained on movies, and investigate three different readouts: trained on movies, trained on noise, and traind on both. We find that when training and testing domain are not the same, there is a clear decrease in prediction performance (Figure 4c, green

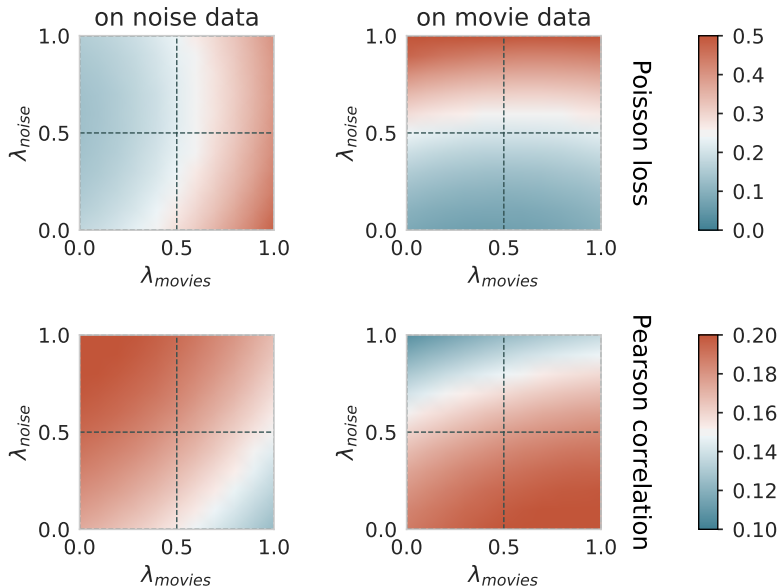

Figure 5: Exploration of the Poisson loss and the correlation between prediction and neural response on the space of readout weights interpolated between the "both" readout and the dedicated readouts (Figure 4a).

and gray bars). This could either mean that the readout weights cannot be correctly learned from the dataset on one domain, or that the real neurons adapt to the particular stimulus domain and no set of readout weights can predict well on both domains. We investigated this question with a readout trained on both stimulus domains. We found that this single readout can perform as well on both domains as each dedicated readout trained on the target domain (compare blue bars to either green or gray). Thus, there is a network that can correctly transfer between both stimulus domains, but that this network cannot be identified by simply training on the natural movie dataset alone.

To corroborate this finding and to determine that difference in performance is caused by the readout weights and not the bias terms we perform a recovery experiment where we linearly interpolate between the weights of the readout trained on both domains and the respective single domain readout weights (controlled by $\lambda_{noise}$ and $\lambda_{movies}$ in Figure 5). For each interpolation, we compute the Poisson loss and the correlation between prediction and neuronal response on the training set. The resulting loss surfaces are consistent with the identifiability hypothesis and show that the network has no gradient information to find the set of readout weights that transfers correctly between both domains (origin in each panel of Figure 5).

## 7   Summary

We presented a novel recurrent network architecture that can fit the responses of thousands of neurons to movies. In addition, our network also accounts for stimulus independent response variation caused by brain state changes and eye movements during the experiment. We demonstrated that both these factors can increase the prediction performance of the network and the ability to transfer neural properties between stimulus domains. To the best of our knowledge, this network is state-of-the-art in predicting neural responses to natural video.

We demonstrated that this network trained on natural movies captures neuronal tuning properties determined on noise. Finally, we showed that there is a network that transfers very well between both stimulus domains, but that this network cannot be identified from training on the natural movie domain alone. One possible avenue to overcome this problem in the future would be to introduce the correct model biases in our architecture via a carefully chosen regularization scheme.

**Acknowledgments**

The authors would like to thank David Klindt and Zhe Li for comments on the manuscript. Supported by the Intelligence Advanced Research Projects Activity (IARPA) via Department of Interior/Interior Business Center (DoI/IBC) contract number D16PC00003. The U.S. Government is authorized to reproduce and distribute reprints for Governmental purposes notwithstanding any copyright annotation thereon. Disclaimer: The views and conclusions contained herein are those of the authors and should not be interpreted as necessarily representing the official policies or endorsements, either expressed or implied, of IARPA, DoI/IBC, or the U.S. Government. Fabian Sinz is supported by the Institutional Strategy of the University of Tübingen (Deutsche Forschungsgemeinschaft, ZUK 63) and the Carl-Zeiss-Stiftung. This work was supported in part by NSF NeuroNex grant 1707400. The authors thank Vathes LLC (`https://vathes.com/`) for hosting the database to reproduce results of this work.

**Statement of Financial Interest**

Edgar Y. Walker, Dimitri Yatsenko, Jacob Reimer, and Andreas S. Tolias hold equity ownership in Vathes LLC which provides consulting for the framework (DataJoint) used to develop/host the data for this publication.

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
