[Supplementary Material]

# Stimulus domain transfer in recurrent models for large scale cortical population prediction on video (Supplementary Material)

**Fabian H. Sinz,**[1-2,5,7,*] **Alexander S. Ecker,**[2,4-6] **Paul G. Fahey,**[1-2] **Edgar Y. Walker,**[1-2]
**Erick Cobos,** [1-2] **Emmanouil Froudarakis,**[1-2] **Dimitri Yatsenko,**[1-2]
**Xaq Pitkow,**[1-3] **Jacob Reimer,**[1-2] **Andreas S. Tolias**[1-4]

[1] Department of Neuroscience, Baylor College of Medicine, Houston, TX, USA
[2] Center for Neuroscience and Artificial Intelligence, Baylor College of Medicine, Houston, TX, USA
[3] Department of Electrical and Computer Engineering, Rice University, Houston, TX, USA
[4] Centre for Integrative Neuroscience, University of Tübingen, Germany
[5] Bernstein Center for Computational Neuroscience, University of Tübingen, Germany
[6] Institute for Theoretical Physics, University of Tübingen, Germany
[7] Institute for Computer Science, University of Tübingen, Germany

[*]`sinz@bcm.edu`

## 1 Natural vs. Rendered Movies

We tested whether there is a difference in the response properties of neurons between natural vs. rendered movies by repeating the domain transfer experiment on a separate scan (`16314-3-1`). While the rendered movie core has a slightly lower overall performance, which can be explained by the fact that there is only 16min of rendered movie time while there is 32min of natural movie stimulus time, natural movie networks transfer to rendered movie networks and vice versa (Fig. S1).

## 2 Natural Image Core vs. Noise Core Control Experiment

One possible confounding factor in our experiments is the difference in presentation time for different stimulus types: 1h natural/rendered vs. 20min noise. This difference could undermine our comparison of the core trained on natural/rendered movies to the core trained on noise movies (Fig. 4 of the main paper). To verify that our conclusions hold, we ran a control experiment where the stimulus time for both conditions was 49 min each. This control experiment produced the same pattern of results, providing independent evidence for our findings presented in Fig. 4 of the main paper.
Overall, the performance of the core trained on noise was slightly higher than in the original experiments, which can be attributed to the larger amount of noise stimuli in this experiment. The two main patterns of results were consistent with the original experiment. First, a core trained on natural movies generalized better to the noise domain (by adapting the readout) than a core trained on noise generalized to the movie domain (Fig. S2A). Second, for the core trained on movies, there existed a common readout for both test domains that performed equally well as the specialized readouts for each individual domain (Fig. S2B).

Figure S1: Control experiment natural vs. rendered movies: Each bar group correspond to one network. Different groups vary in the training conditions, i.e. the type of data used for the core or the readout. The difference between the bars in each group is the difference in test performance for the rendered vs. natural movie domain. The fact that the test performances matches very well indicates that networks transfer well between both domains.

(a) Cross-domain performance for the noise and natural movie core.

(b) Performance of readout is domain dependent.

Figure S2: Control experiment with balanced presentation time. Panels (a) and (b) replicate panels (b) and (c) of Fig. 4 in the main paper, respectively. Note that the neurons underlying this plot are from a different scan 16314-3-1 and the absolute scales are not comparable to the scores in the main paper .