[Reviews · NeurIPS 2018]

Reviewer 1



##### I have read over the author's rebuttal, and I am satisfied by their clarification and response. ##### In this submission, the authors use modern deep learning techniques to develop a predictive model of mouse V1 cortex. Most notably, this model seeks to fill a gap in the literature for simultaneously predicting large populations of neurons responding to both natural video and noise stimuli, whereas previous models had mainly focused on static image stimuli in small cortical populations. The model developed by the authors receives these video stimuli as inputs, and consists of well-established computer vision architectures — here a feed-forward convolutional network feeding into a convolutional GRU — to extract stimulus feature representations common to all neurons. These are then read-off and modulated by a spatial-transformer network to ensure receptive-field stability of the regressed neural outputs. A novel structural contribution has been added to account for non-stimulus driven experimental noise which might be present in the data. Specifically, the readouts are additionally modified by “shift” and “modulator” networks which factor in these sources of noise such as pupil movement (correction to this is needed for proper receptive-field mapping), pupil diameter, and measured running speed of the animal (which is known to be correlated with V1 activity in rodents). They show that this model outperforms a non-recurrent linear-nonlinear control model (fig. 2a) across both datasets for all three mice, and assess the relative contributions of the novel shift and modulator component networks (fig. 2c). In “domain transfer” experiments, they show both qualitative and quantitative evidence for models generalizing outside their training set to produce receptive fields and tuning curves similar to mouse V1 activity, even though the mice have been shown different noise stimuli. Lastly, the authors study the quantitative amount of transfer between stimulus domains and find that models trained on movies serve as reasonable predictors of white-noise induced activity, and find that the converse is not true (fig. 4b). However, a network which combined domain knowledge from both stimuli sets provides a sufficient basis space for explaining activity presented in either domain (fig. 4c). Overall, I am impressed by the ambition of this work. I believe that the field could benefit from more of this style of large-scale modeling, and further consideration of generalization to novel stimuli as a complex yet intriguing phenomenon. The application to large-scale populations responding to complex stimuli, instead of only a small subset of V1 responding to simpler stimuli is also noteworthy. I found the introduction of the shifter and modulator networks to be a novel idea, encoding well-thought-out structural priors into the model. That being said, although I have no complaints concerning the high-level scope of analysis presented in this submission, I do have some concerns as to the strength of results for several reasons: (1). I am willing to believe the qualitative comparison between the “full” model to a “standard” linear-nonlinear system in fig 2a. Based on the lack of recurrence alone, the result seems intuitive. However, I am concerned with the quantitative estimate between these two models due to not controlling for numbers of learnable parameters. From what I understand, the authors balance the number of parameters in the ConvNet betweeen models — three 12-feature layers in the full model and one 36-feature layer in the LN mode — and readout structures. However, in the absence of the GRU in the LN model, there is an imbalance in these numbers. Generally speaking, you can interpret this in two ways: either (i). the extra units and nonlinearities can artificially inflate the score of the full model if no hyperparameter optimization has been performed or (ii) the authors had performed a hyper parameter sweep (number of units) across both models and settled on the best performing settings. Given that no mention towards the latter has been discussed in the submission, I am inclined to believe that the former is true. (2). I am concerned with a statement that was made inside the “Network Implementation” subsection which stated that all readouts biases were initialized to the mean firing rate of that neuron. It seems to me that this choice is also reasonable, however it sounds like that is a strong prior for what these values should be. Could the authors discuss the impact of this initialization, and how it relates to why the full model might only be performing ~50-70% of the oracle estimator? (3). Similar to the above in (1), even if I am intrigued by the application of the shift and modulator networks, I am concerned that for parameter balancing reasons and the lack of a hyperparameter sweep, the results in 2c might be artificially inflated. Additionally, I’m concerned that the novelty of such an approach might be detracting from what might be a more standard way of incorporating these stimulus-independent noise sources — namely, by having them as generic features that are concatenated as inputs to the readout MLP, rather than these specific priors. I would be just as impressed by seeing a larger/deeper readout network with a more generic structure perform at the level (or better) than something with a “clever” implementation. (4). Almost unsurprisingly, the authors find that models trained on movies are well-correlated with noise-induced activity, although the converse is not true, and that a model trained on both can in fact predict both. I would have liked a similar comparison between natural movies and the rendered videos as well, because it is unclear whether the statistics between even these two domains are imminently transferable. What I did find interesting here is that although the movie-optimized models are well correlated with noise responses, they show high variance in predicting the low-levels details such as orientation and direction tuning. The paper ends on a bit of a cliff-hanger in the sense that they do not propose a better method of a generalized model than what amounts to just training on all stimulus contingencies. In summary, I am inclined to like this work. The results — although not necessarily as strong as they could be — feel intuitive. If text length were not a factor, I personally feel like I would have benefitted from an extended discussion/interpretation section of results, and more detailed analysis of those presented in the submission (specifically as far as the domain transfer section is concerned). The approach and aims of this work alone are a worthwhile contribution to the neuroscience community as a whole.

Reviewer 2



EDIT: I have read the author's rebuttal and found it to provide convincing answers to questions about the performance and value of their model. Their approach of "(i) build good predictive models, (ii) explore them in-silico to generate experimental predictions where we are not constrained by experimental time and (iii) test these predictions in-vivo." is in my opinion about as good as any in computational neuroscience at the moment. They don't make the tenuous claim that their model is explanatory in virtue of it's predictive power. Summary and relation to previous work: The submission presents a new deep recurrent architecture to predict neural activity of thousands of mouse V1 neurons, recorded simultaneously with two-photon microscopy. This architecture allows for the prediction of responses to natural video, a more ecologically valid stimulus than commonly used paramterized noise, while taking into account potentially confounding variables that are not related to the stimulus. The authors investigate whether a model trained on natural video can generalize to responses to noise and vice versa. They find that finetuning the last layer on the target domain leads to successful transfer from video to noise, but that the converse is not true. Quality: The authors carefully compare the performance of their model to an oracle estimator as an estimated upper bound on the best performance possible. The authors rigorously perform a number of experiments that provide confidence that the filters they found are true properties of the neurons and not artifacts of the data, model or training. In particular, the successful domain transfer from natural video to noise but not from noise to natural video strongly justifies the use of natural videos over controlled stimuli such as parameterized noise. Clarity: The writing is clear and well organized. Sufficient details are provided to reproduce the results. However, the data does not appear to be publicly available nor do the authors share their code. Originality: To the best of my knowledge, predicting neural responses at this scale (several thousand simultaneous recordings) to natural movies in awake mice, while taking into account confounding animal state variables, is a relatively new task with no successful models. Previous work has focused on subsets of the challenges tackled here. Significance: This submission contributes to a much needed effort to develop new methods to perform system identification on thousands of neurons recorded simultaneously. The model is especially significant because it models responses in more ecologically valid experiments involving awake and moving animals watching natural videos, while taking into account confounding variables just as pupil position and animal speed. This work also adds to the growing body of work showing that V1 filters are much more diverse than the traditionally assumed Gabor filters. Several aspects of this work could be built upon in future work in computational neuroscience. Minor edits: - There should be a newline before the beginning of paragraphs: line 132, 208, 231, 257, 263, 286 - line 16: ungrammatical, replace "generalizing" with "generalize"

Reviewer 3



######### The rebuttal clarifies some of the technical issues. In particular I find the clarification about the network performance to be helpful. I’ve increased my score from 5 to 6. However, the issue of temporal correlations in the calcium traces is not addressed in the rebuttal. I still wonder how much the residual correlation in the calcium input has affected the results here. In particular, I am still worried that the model is in fact capturing at least part of the temporal dynamics in the calcium measurement, not the actual spiking responses. ######### The manuscript describes a models based on deep network architecture to fit the inferred spiking activity of mouse V1 neurons recorded using 2p calcium imaging. The authors fit the proposed model to the actual neural responses from mice and evaluate the model performance for various scenarios. They also studied whether the model could transfer between domains, based on artificial noise stimuli and natural images. I like the general approach the authors are taken. The model architecture the authors have put together (including Core+Readout+Shifter+Modulator) seems to be novel and reasonable. And, not too many previous models have been able to deal with the video data. My main concerns with this manuscript are about what one actually learns when applying this sophisticated model to the real data. Overall, I consider this is a decent method paper, but at the same time, the applications to the real neural data need to be further developed to make it a stronger submission. In particular- * The model performance shown in Fig2 is not particularly impressive. Fig2b is a bit misleading, because as stated in the text, the oracle model represents an upper bound when only stimulus is consider (but not the behavioral state). I am a bit puzzled about the relatively low performance of this model. Is it because of the high noise/low temporal resolution in the calcium imaging data or with the model itself (could be missing a key component)? * The authors demonstrate that tuning curves can be inferred from this model. However, how is this new approach advantageous over to standard GLM approach in this regard. Are there additional insights could be revealed using this more sophisticated approach? * In the last Result section, it is shown that feature representation based on natural movie transfer to noise stimuli, while the converse is not sure. While it is certainly re-assuring to see this, it is also what one would naturally expect. There is a large theoretical and experimental literature on studying how the neural representation adapt to the statistical structure of the environment, in particular the structure of the natural images Thus, I am still unsure what this reveals. * How did the authors make sure that the temporal correlations in the calcium traces have been removed? It seems important to get rid of the the correlation induced by the calcium dynamics here. If there is “residual” temporal correlation in the deconvolve “spiking” activity, it might affect the fitting of the model. It is possible that this might be part of the reason why the model performance is not particularly good. Alternatively, it could be that the model is in fact capturing the temporal dynamics in the calcium measurement, not the actual spiking responses.